# Real-world effects of medications for stroke prevention in atrial fibrillation: protocol for a UK population-based non-interventional cohort study with validation against randomised trial results

Emma Maud Powell ,[1] Ian J Douglas,[1] Usha Gungabissoon ,[2] Liam Smeeth,[1] Kevin Wing[1]

[1]Department of Non-communicable Disease Epidemiology, Faculty of Epidemiology and Population Health, London School of Hygiene and Tropical Medicine, London, UK
[2]Epidemiology (Value Evidence and Outcomes), GSK, London, UK

**Correspondence to**
Emma Maud Powell;
maud.teoh@lshtm.ac.uk

## ABSTRACT

**Introduction** Patients with atrial fibrillation experience an irregular heart rate and have an increased risk of stroke; prophylactic treatment with anticoagulation medication reduces this risk. Direct-acting oral anticoagulants (DOACs) have been approved providing an alternative to vitamin K antagonists such as warfarin. There is interest from regulatory bodies on the effectiveness of medications in routine clinical practice; however, uncertainty remains regarding the suitability of non-interventional data for answering questions on drug effectiveness and on the most suitable methods to be used. In this study, we will use data from Apixaban for Reduction in Stroke and Other Thromboembolic Events in Atrial Fibrillation (ARISTOTLE)—the pivotal trial for the DOAC apixaban—to validate non-interventional methods for assessing treatment effectiveness of anticoagulants. These methods could then be applied to analyse treatment effectiveness in people excluded from or under-represented in ARISTOTLE.

**Methods and analysis** Patient characteristics from ARISTOTLE will be used to select a cohort of patients with similar baseline characteristics from two UK electronic health record (EHR) databases, Clinical Practice Research Datalink Gold and Aurum (between 1 January 2013 and 31 July 2019). Methods such as propensity score matching and coarsened exact matching will be explored in matching between EHR treatment groups to determine the optimal method of obtaining a balanced cohort.

Absolute and relative risk of outcomes in the EHR trial-analogous cohort will be calculated and compared with the ARISTOTLE results; if results are deemed compatible the methods used for matching EHR treatment groups can then be used to examine drug effectiveness over a longer duration of exposure and in special patient groups of interest not studied in the trial.

**Ethics and dissemination** The study has been approved by the Independent Scientific Advisory

### Strengths and limitations of this study

► Selection of electronic health record patients matched to the randomised controlled trial (RCT) patients allows assessment of the ability of non-interventional methods to detect effectiveness of treatments for stroke prevention in atrial fibrillation (AF) within an RCT-analogous population.

► Combined Clinical Practice Research Datalink (CPRD) Gold and Aurum population broadly representative of the patients prescribed apixaban and warfarin for AF in routine clinical practice in the UK.

► Some of the criteria that were assessed for ARISTOTLE eligibility may not be well recorded in CPRD.

► Adherence to medication will need to be assessed based on proxy variables (time covered by prescription for the direct-acting oral anticoagulants, time in therapeutic range based on international normalised ratio measurements for warfarin); the different nature of these proxy variables means the adherence estimates may not be comparable.

► Ascertainment of outcomes via CPRD is based on recording as part of routine clinical care rather than for specifically detecting study outcomes.

Committee of the UK Medicines and Healthcare Products Regulatory Agency. Results will be disseminated in scientific publications and at relevant conferences.

## INTRODUCTION
### Background and rationale

Atrial fibrillation (AF) is a common cause of cardiac arrhythmia with symptoms including palpitations, fainting and shortness of breath; however, some patients may be asymptomatic. The prevalence of AF in the UK is estimated to be around 3%,[1] increasing from 0.2% in

people aged 45–54 years to 8.0% in those 75 and older.[2] The lack of organised atrial contraction in AF can lead to the formation of thrombi, meaning that patients with AF have a fivefold higher risk of stroke which is an important cause of morbidity and mortality.[3–5]

Current UK guidelines recommend use of prophylactic treatment with anticoagulation medication to reduce the risk of stroke. Warfarin, a vitamin K antagonist (VKA) and the previous standard anticoagulant treatment, has many treatment and dietary interactions requiring frequent monitoring of a patient's international normalised ratio (INR), to maintain anticoagulant activity within a narrow range (2.0–3.0). Low levels put the patient at a higher risk of stroke while high levels lead to a higher risk of bleeding.[6] In 2011, the first direct-acting oral anticoagulant (DOAC) dabigatran was approved for the treatment of AF in the European Union (EU); it was anticipated to provide easier to manage long-term anticoagulation therapy for patients with AF given the complex safety profile of warfarin. ARISTOTLE, a pivotal randomised controlled trial (RCT) of the DOAC apixaban, demonstrated superiority over warfarin for both prevention of stroke and safety (major bleeding) among individuals with AF.[7]

The generalisability of the ARISTOTLE trial is limited by the strict eligibility criteria; evidence on apixaban's treatment effect is therefore lacking for patients who would not have met the eligibility criteria such as those at increased bleeding risk or with severe comorbid conditions. The regulatory environment now demands evidence of treatment effectiveness outside the confines of randomised trials.[8 9] Non-interventional data sources have the potential to overcome many of the RCT limitations given that they contain data for a wide spectrum of patients treated with the drug in routine care, including patients who would have been not eligible for trials. Data collected as part of routine patient care such as electronic health record (EHR) provide a valuable opportunity to obtain evidence on the effectiveness of apixaban in a routine care setting. A key problem with using these data is that the absence of randomisation leaves them highly susceptible to confounding making it difficult to have confidence in the results.

To address this lack of confidence, this study will apply innovative matching approaches to create a trial-analogous non-interventional cohort for analysis. Records from UK EHRs will be matched to ARISTOTLE patients before using methods for matching between treatment groups within the non-interventional EHR data, creating an EHR population similar to the trial population that is well balanced by treatment group. If successful, estimates of effectiveness and safety of apixaban obtained from analysis of this ARISTOTLE-analogous cohort should be comparable with the results from the ARISTOTLE trial. The non-interventional analysis methods used to obtain these results may then be used to reliably estimate effects in understudied AF patient groups.

## AIMS AND OBJECTIVES

The aims of this study are (1) to measure the association between anticoagulation treatments for stroke prevention in AF and time to stroke, systemic embolism (SE), myocardial infarction (MI), major bleeding and mortality among an ARISTOTLE-analogous cohort of patients from UK EHRs, and (2) to develop a methodological framework with in-built validation for using observational EHRs to answer questions about DOAC risks and benefits in patients not included or under-represented in the RCTs.

The specific objectives are to:

Objective 1. Check comparability of EHR data and robustness of methods for measuring stroke prevention medication effectiveness in an ARISTOTLE-analogous cohort using data from EHR data and by comparing with ARISTOTLE results.

Objective 2. Extension of trial findings: measure treatment effects of apixaban in patient groups excluded from ARISTOTLE.

Objective 3. Comparative effectiveness: compare treatment effectiveness between multiple individual anticoagulants (warfarin, apixaban, rivaroxaban, dabigatran) in ARISTOTLE-eligible cohorts and in patient groups excluded from ARISTOTLE.

## METHODS AND ANALYSIS

Figure 1 (figure adapted from a study in real-world effects of medications for chronic obstructive pulmonary disease[10]) provides an overview of the study, covering the objectives and data sources used, and how RCT data will be used in Objective 1 to validate methods for analysing effectiveness of treatments for stroke prevention in AF in non-interventional data. Should Objective 1 prove successful the validated methods will be applied to unanswered questions in Objectives 2 and 3.

### Study design

We will use a retrospective cohort study design using longitudinal data to evaluate the effects of prescribing apixaban versus warfarin and then versus other DOACs for prevention of stroke and SE in AF on key effectiveness and safety outcomes using non-interventional primary care data.

### Setting/data sources

Patient data used in this study will be obtained from several sources: primary care data on UK National Health Service (NHS) patients from Clinical Practice Research Datalink (CPRD) Gold and Aurum databases, additional data on hospital events and mortality on UK NHS patients with linked data from the Hospital Episodes Statistics (HES) and Office for National Statistics (ONS) databases, and results from the ARISTOTLE trial.

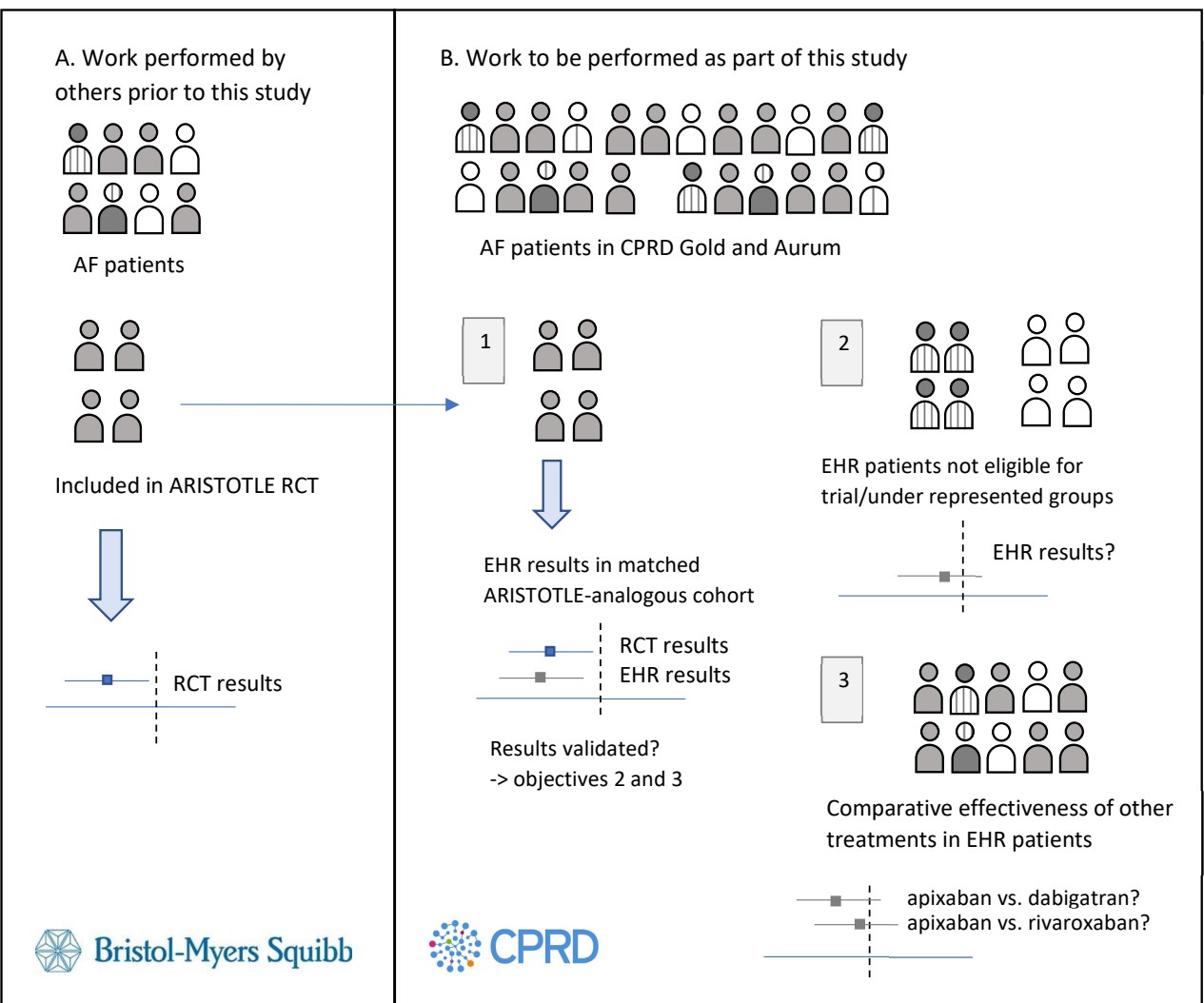

**Figure 1** Overview of study objectives and sources of data for the real-world effects of medications for stroke prevention in AF study. AF, atrial fibrillation; CPRD, Clinical Practice Research Datalink; EHR, electronic health record; RCT, randomised controlled trial. (A) Work performed by others prior to this study. ARISTOTLE: RCT that investigated effectiveness and safety of apixaban vs warfarin in prevention of stroke and systemic embolism in AF patients. RCTs results inform clinical practice despite only a subset (based on trial inclusion and exclusion criteria) of the total population of AF patients being included in the RCTs of stroke prophylaxis treatments. (B) Work to be performed as part of this study. (1) Objective 1. A cohort of ARISTOTLE-analogous patients will be selected from UK EHRs (CPRD Gold and Aurum), by matching EHR patients prescribed apixaban to the apixaban patients included in the trial on baseline characteristics. EHR patients prescribed warfarin will then be matched to the trial-analogous EHR apixaban patients. An analysis of the effectiveness of apixaban vs. warfarin on prevention of stroke/ systemic embolism will then be performed on this ARISTOTLE-analogous EHR cohort. If the results obtained are comparable to those obtained in ARISTOTLE, this will serve as a validation step, showing that data from the non-interventional CPRD Gold and Aurum sources can reliably be used to study stroke prevention treatment effects in AF. (2) Objective 2. The validated analysis techniques used for Objective 1 will then be used to study UK EHR patients who would not have been eligible for inclusion in an RCT or are under-represented in RCTs due to their age or presence of other comorbidities, for whom the comparative effects of anticoagulants in stroke prevention in AF is unclear.(3) Objective 3. The validated analysis techniques used for Objective 1 will then be used to compare effectiveness of apixaban vs warfarin, apixaban vs rivaroxaban and apixaban vs dabigatran.

## ARISTOTLE

ARISTOTLE was a randomised, double-blind trial completed in 2011, comparing apixaban with warfarin in the prevention of stroke and SE. The trial included 18 201 patients with AF and at least one additional risk factor for stroke. The trial was designed to test for non-inferiority of apixaban compared with warfarin, and showed apixaban superiority for (1) the primary outcome of stroke or SE (HR 0.79, 95% CI 0.66 to 0.95),[7] (2) the safety endpoint of major bleeding (HR 0.69, 95% CI 0.60 to 0.80), and (3) death from any cause (HR 0.89, 95% CI 0.80 to 0.99). The ARISTOTLE findings led to the National Institute for Health and Care Excellence (NICE) guidelines on stroke prophylaxis in patients with AF recommending apixaban as a treatment. Baseline patient characteristics

from ARISTOTLE will be used in selection of participants in Objective 1.

## CPRD Gold

CPRD Gold is a database containing anonymised data from over 625 primary care practices across the UK (approximately 13 million patient records) and is representative of the UK population with respect to age, gender and ethnicity.[11] Gold contains information on clinical diagnoses, prescribing, referrals, tests and demographic/lifestyle factors. General practices must meet prespecified standards for research-quality data to contribute data.

## CPRD Aurum

CPRD Aurum contains primary care records similar to Gold but based on practices using EMIS software, whereas Gold has data from practices using Vision software. CPRD Aurum contains data on 19 million patients from 738 practices (10% of English practices) with 7 million active patients.[12]

## Selection of participants

Participants will be selected from CPRD Gold and Aurum between 1 January 2013 and 31 July 2019. All patients will need to have been registered with a practice contributing

research quality data for at least 6 months. Participant selection criteria will then vary by objective as detailed below.

### Objective 1

An overview of each of the steps for participant selection for Objective 1 is provided in figure 2.

### *Step 1*

We will select all (HES and ONS linked) patients in the EHR cohort (CPRD Gold and Aurum) who would have met the following *inclusion* criteria for the ARISTOTLE study, at least 6 months after patient registration in the database on or prior to the index date:

► Diagnosis of AF.
► Age 18+ years.
► One or more stroke risk factors (age 75 years or older; prior stroke, transient ischaemic attack or SE; congestive heart failure; diabetes mellitus; hypertension).

In ARISTOTLE, patients randomised to apixaban were new users of apixaban while both treatment arms were allowed to be previous users of warfarin, with patients stratified by prior warfarin/VKA exposure. To mirror ARISTOTLE, we will assess trial criteria for apixaban

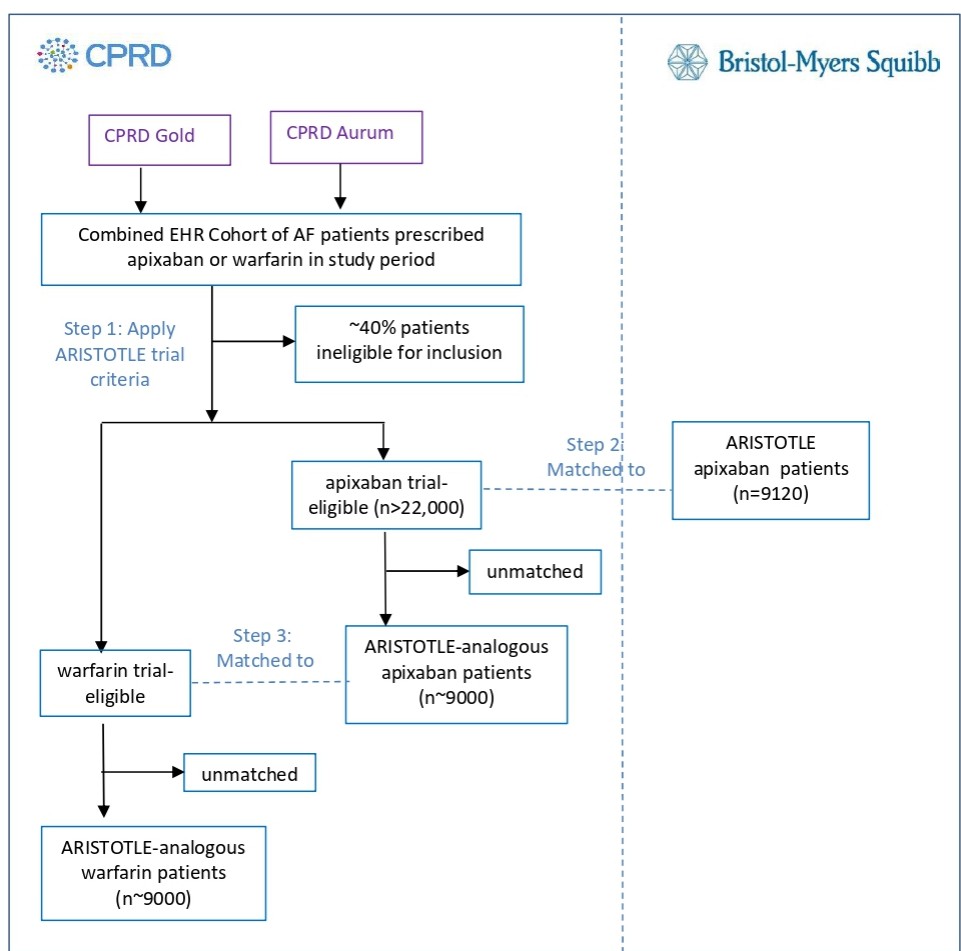

**Figure 2** Flow chart illustrating the assembly of a matched trial-analogous cohort of EHR patients. AF, atrial fibrillation; CPRD, Clinical Practice Research Datalink; EHR, electronic health record.

patients on the date of their first prescription of apixaban while allowing patients prescribed warfarin to become eligible at any warfarin prescription date during the study period; furthermore, we will match ARISTOTLE in the proportion of new versus prevalent users in both treatment arms. We will then exclude patients who meet any of the following ARISTOTLE study *exclusion* criteria prior to their eligible-for-inclusion date:

▶ AF due to reversible causes.
▶ Mitral stenosis.
▶ Increased bleeding risk.
▶ Conditions other than AF requiring chronic anticoagulation.
▶ Persistent, uncontrolled hypertension.
▶ Active infective endocarditis.
▶ Current treatment with aspirin >165 mg/day.
▶ Simultaneous current treatment with both aspirin and a thienopyridine.
▶ Conditions likely to interfere with participation in the trial or cause death within 1 year.
▶ Recent alcohol or drug abuse, or psychosocial reasons making study participation impractical.
▶ Recent ischaemic stroke (within 7 days).
▶ Severe renal insufficiency.
▶ Alanine aminotransferase or aspartate aminotransferase >2× upper limit of normal (ULN) or total bilirubin ≥1.5× ULN.
▶ Platelet count ≤$100 \times 10^9$/L
▶ Haemoglobin <90 g/L.
▶ Pregnancy or breast feeding.

Feasibility counts in Gold found approximately 60% of patients with AF prescribed apixaban met the ARISTOTLE trial criteria. Details of the algorithms used in applying the trial criteria to the EHR data are given in the online supplemental file.

### Step 2
We will select a subset of apixaban patients from our EHR pool to create a cohort that matches the ARISTOTLE apixaban participants on a selection of the following baseline characteristics:

▶ Age.
▶ Sex.
▶ Body mass index (BMI).
▶ Systolic blood pressure (SBP).
▶ Congestive heart failure or left ventricular systolic dysfunction.
▶ Hypertension requiring treatment.
▶ Diabetes mellitus.
▶ Prior stroke/thromboembolism.
▶ Smoking status.
▶ Alcohol consumption.
▶ Level of renal impairment.
▶ Prior VKA/warfarin exposure.
▶ Labile INR in prior users of warfarin.
▶ Concomitant use of: aspirin, antiplatelet or non-steroidal anti-inflammatory drug, lipid-lowering drug therapy, or CYP3A4 inhibitor.

This step will generate a group of ARISTOTLE-analogous apixaban patients, with similar baseline characteristics to ARISTOTLE subjects at the point of randomisation (n~9000).

The variables selected are expected to influence the likelihood of the outcomes of interest. Exact selection of matching variables will depend on the quality and completeness of the data available and a balance will be struck between matched sample size and balance. Different methods to facilitate selection of a matched cohort will be explored, such as propensity score matching (PSM) and coarsened exact matching (CEM),[13] a non-parametric method that may give estimates with lower variance and bias for a given sample size compared than other methods.[14]

### Step 3
The resulting trial matched sample of EHR apixaban patients will be matched to the warfarin ARISTOTLE-eligible EHR patients (figure 2) using a matching method such as PSM or CEM (final method selected based on giving optimal sample size vs balance). Risk set sampling will be employed in order to ensure similar duration of prior VKA/warfarin exposure for the prevalent users in the apixaban and warfarin EHR cohorts. The covariates for consideration in matching between EHR treatment arms or construction of a propensity score (PS) model will include the variables listed in step 2 along with additional EHR variables such as data source (Gold or Aurum), socioeconomic status and comorbidities. Each apixaban patient from the ARISTOTLE-eligible EHR patients will be matched 1:1 with the warfarin EHR patient with the closest match giving a trial-analogous cohort of ~18 000.

### Step 4
The absolute rates and HR for the outcomes of interest (time to: stroke/SE, MI, major bleeding and mortality) will then be calculated. For the primary outcome (time to stroke/SE) the EHR results will be validated against the ARISTOTLE trial results using the criteria detailed in the Statistical Analysis section (Validation of observational results against ARISTOTLE data).

### Objective 2
We will select patient groups who would not have been included in ARISTOTLE (and therefore would not have been included in the Objective 1 cohort) or who are under-represented in ARISTOTLE. Specifically, this will include patient groups such as patients with an AF diagnosis in the EHR cohort meeting these additional criteria:

▶ Severe comorbid condition: disease with a likelihood of causing death within 1 year or reasons making participation unpractical (such as dementia).

When matching the apixaban and warfarin patients within the patient groups for this objective, additional baseline variables will be considered compared with the list specified for Objective 1, Step 2; namely the H, A, and B components of the HAS-BLED score (Hypertension, Abnormal renal/liver

function, Stroke, Bleeding history or predisposition, Labile INR, Elderly (>65 years), Drugs/alcohol concomitantly) not included for Objective 1 matching due to being ARISTOTLE exclusion criteria. In these special patient populations the same outcomes as Objective 1 will be assessed, with absolute and relative rates calculated separately in each special patient group.

### Objective 3

We will select all patients with AF who have a prescription for apixaban, warfarin, rivaroxaban or dabigatran in the treatment period (between 1 January 2013 and 31 July 2019). The ARISTOTLE trial criteria will be applied, followed by matching the warfarin, rivaroxaban and dabigatran ARISTOTLE-eligible EHR patients in turn to the trial-eligible EHR apixaban patients following the methodology outlined in Objective 1, Step 3. This process will result in the creation of three trial-eligible EHR cohorts: warfarin users matched to apixaban users, rivaroxaban users matched to apixaban users and dabigatran users matched to apixaban users. Matched cohorts of excluded patient groups will also be constructed to enable pairwise comparisons of treatment effects in these groups using the method outlined in Objective 2. In all cohorts, the same outcomes as Objective 1 will be assessed with both absolute and relative treatment effects compared.

### Exposures, outcomes and covariates

#### Exposures

For all objectives, exposures will be determined using CPRD Gold and Aurum prescribing records and code lists for anti-coagulant treatments with no restrictions placed on the dose prescribed.

For Objectives 1 and 2, use of apixaban is the primary exposure of interest and will be compared with warfarin.

For Objective 3, other stroke prevention treatments for AF will also be compared, namely dabigatran and rivaroxaban.

#### Outcomes

Outcomes to be measured are as follows:
► Stroke (ischaemic or haemorrhagic) or SE.
► Major bleeding.
► MI.
► All-cause mortality.
► Time to AF treatment change.

Outcomes will be ascertained using a combination of CPRD, HES and ONS data.

#### Covariates

The variables to be considered for matching patients are detailed in the selection of participants for Objective 1 (Step 2).

### Sample size

#### Objective 1

ARISTOTLE included 9120 patients in the apixaban arm, therefore it was estimated a minimum of 15 000 EHR apixaban patients were needed for matching to be feasible. In CPRD Gold, approximately 8400 patients were eligible (January 2018). Aurum (June 2019) contained 23 526

apixaban patients with AF not registered in practices that had previously contributed data to Gold. Assuming the proportion of Aurum patients meeting ARISTOTLE eligibility criteria would be similar to the proportion in Gold (~60%) gave an estimate of 14 115 trial-eligible apixaban patients. Combining Gold and Aurum is therefore estimated to give >22 000 unique trial-eligible EHR apixaban patients.

#### Objectives 2 and 3

From feasibility counts, we are confident we will have sufficient numbers of patients to allow well-powered analyses for Objectives 2 and 3. For example, we estimate the number of people with no evidence of at least one additional risk factor for stroke for Objective 2 would be >3000 people in each exposure group.

### Statistical analysis

#### Methods of analysis

ARISTOTLE used an intent-to-treat (ITT) approach for the primary efficacy analysis and an on-treatment approach for sensitivity analysis and safety outcomes. We will perform equivalent analyses by using two different censoring schemes: a primary censoring scheme censoring 5 years after index date (reflecting the maximum possible follow-up in ARISTOTLE) for the primary effectiveness analyses, and an on-treatment scheme censoring around time of last study drug for the sensitivity analysis and safety outcome. For the on-treatment censoring scheme, date of last exposure will be estimated using patient prescription data—to allow for drug half life, stockpiling of tablets and less than 100% adherence we will add 30 days after the apparent end of treatment.

Demographic and baseline variables will be presented before and after matching steps. As the primary analysis accounts neither for treatment switching nor discontinuation, the proportion of patients discontinuing treatment and time to treatment discontinuation will be tabulated.

The primary effectiveness endpoint is time to first occurrence of confirmed stroke (ischaemic, haemorrhagic or unspecified type) or SE during the study, regardless of whether the subject is receiving treatment at the time (primary censoring scheme). Comparisons will be made according to prescribed treatment (apixaban vs warfarin).

All time to event endpoints will be analysed using a Cox proportional hazards model including treatment group as a covariate and prior warfarin/VKA status (experienced, naïve). Point estimates and two-sided 95% CIs will be constructed for the outcome. Absolute event rates of all outcomes of interest will also be calculated.

Secondary outcomes cover the key safety outcome of major bleeding and the individual outcomes of stroke, SE, MI and mortality. Secondary outcomes other than major bleeding will use the ITT censoring scheme, major bleeding will use the on-treatment censoring scheme.

### Validation of observational results against ARISTOTLE data

In Objective 1 alone, we will validate the findings from our primary analysis against ARISTOTLE by determining

whether results are compatible with the trial results. ARISTOTLE demonstrated superiority of apixaban over warfarin for the primary endpoint (HR 0.79, 95% CI 0.66 to 0.95).[7] The treatment effect seen with EHR data may be weaker than that seen in ARISTOTLE.

An analysis of EU patients in ARISTOTLE showed a smaller treatment difference for the primary endpoint and death: HR for stroke/SE 0.92 (95% CI 0.56 to 1.52), all-cause death 0.89 (95% CI 0.68 to 1.18). The European Medicines Agency Assessment Report suggested the smaller treatment effect may have been due to superior INR control in the warfarin arm of the EU subgroup (median time in therapeutic range (TTR) 68.93%)[15]; this study could provide additional evidence on this point.

Either a result of superiority or non-inferiority will be considered compatible with ARISTOTLE results. We have set two criteria that must be met to conclude results are consistent with the trial result:

1. The effect size must be clinically comparable with the ARISTOTLE findings; the HR for time to stroke/SE with the EHR must be between 0.69 and 0.99. This range is not symmetrical around the ARISTOTLE estimate of 0.79 as it is anticipated the treatment effect in routine clinical care may be weaker than that seen in the optimised setting of a clinical trial.
2. The upper limit of the 95% CI for the rate ratio must be less than 1.52 (upper limit in the EU subgroup of ARISTOTLE).

In addition, if the upper limit of the 95% CI is less than 1 then superiority of apixaban versus warfarin will be concluded.

In order to understand the extent to which the EHR population resembles the ARISTOTLE trial population the absolute event rates of the outcomes will be compared.

### Sensitivity analyses

Primary and secondary effectiveness outcomes will also be analysed using the on-treatment censoring scheme to investigate whether the extent of treatment discontinuation compromises confidence in the effectiveness analyses.

Exclusion of patient time post-treatment discontinuation in the safety and sensitivity analyses might bias results towards a conclusion of no difference[16] and risks selection bias due to attrition[17]; the set of patients who switch or discontinue treatment will therefore be examined to ascertain whether biases of this nature may have occurred.

Additional analyses may be performed using methods such as inverse probability of censoring weighting (IPW) or a rank-preserving structural failure time model to estimate the treatment effect that would have been observed in the absence of treatment switching. We will explore the impact of time-varying eligibility by using methods such as a modified treatment strategy IPW.[17]

Adherence will be estimated in the EHR cohort to enable comparisons with the trial and investigate the extent to which this may have influenced differences in treatment effect observed. For apixaban, we will calculate the proportion of days covered (PDC) over a patient's time when on treatment

as a measure of adherence. Warfarin dose is poorly recorded in EHR, therefore warfarin adherence will be estimated by looking at adherence to other long-term daily medications as a proxy measure and by looking at INR control by calculating per cent INR TTR as a measure of overall warfarin treatment regime adherence.

We will perform a supplementary analysis in patients deemed adherent (PDC ≥80% matching ARISTOTLE compliance limit) along with an exploratory subgroup analysis by INR TTR. The different nature of the proxy variables used for adherence in the DOACs (PDC) compared with warfarin (INR TTR) means that the adherence estimates may not be comparable; should great differences in adherence be observed between these exposure groups the definitions of adherence used may need to be reassessed.

Apixaban was a newly available drug with a low number of patients having a prescription in the first year it was available[18]; we will therefore perform a sensitivity analysis with the start of the study period shifted forwards a year to January 2014 to investigate the impact of inclusion of early adopters who may differ from later adopters of a new drug.

### Plan for addressing confounding

In the study period, apixaban was a newly available treatment leading to the possibility of channelling bias. For Objective 1, by applying trial eligibility criteria to both treatment cohorts and matching using the baseline covariates we should avoid channelling bias. To handle confounding, treatment arms will be matched using the optimal method selected. Unmeasured or unknown confounding may remain and this will be explored in the analysis and discussion of results.

### Missing baseline data

UK EHR data have been shown to be almost complete for drug prescribing and information on important comorbidity is well recorded. For some variables such as renal function and alcohol intake, a patient is more likely to have no data entered if there is no overt clinical evidence of abnormality; in such cases, we may take a pragmatic approach categorising into a parameter ('evidence of' vs 'no evidence of') with those with no data included in the 'no evidence of' group. For BMI and SBP, we cannot assume data are missing at random as we expect a patient is less likely to have these recorded if they appear at a healthy weight and do not have hypertension, respectively, or if they have a lower comorbidity burden. Furthermore, as the proportion of patients with missing baseline BMI or SBP is expected to be low (approximately 4% for BMI and <1% for SBP[18]), these patients will be excluded from the trial-eligible cohort.

### Missing prescription data

Treatment may be initiated in secondary care, meaning the first prescription of patients newly initiating treatment or switching treatments is missing; to account for this we will perform a sensitivity analysis where those newly initiating treatment are assigned an earlier derived index date. Hospitalised patients may have prescriptions in secondary care leading to treatment gaps in their primary care data.

We will investigate the occurrence of hospitalisation around treatment discontinuation and assess the potential impact on the results of missed events by performing a sensitivity analysis with different extended derived dates of last dose. Some concomitant drugs used in determining eligibility and matching patients are available over the counter (OTC), meaning we may miss that patients are exposed to these; we expect OTC use of these drugs to be similar in both treatment groups.

### Missing outcome data

EHR data are shown to be almost complete for mortality.[19] Patient deaths missing from EHRs are expected to be missing at random equally in both treatment arms, thereby not altering the overall direction of treatment effect. The classification of unspecified stroke type will cause uncertainty in the main safety endpoint and may lead to a lower event rate for major bleeding compared with the trial; this would affect the power but should not affect the treatment effect seen as events are expected to be missing at random from both treatment arms.

### Limitations of the study design, data sources and analytical methods

Some of the criteria assessed for ARISTOTLE eligibility may not be well recorded in CPRD, criteria such as alcohol and drug abuse may not be captured for all patients. For criteria such as 'increased bleeding risk', it is unclear which codes to include and timescale to consider. These limitations are consistent with our aim to select a population as similar as possible to ARISTOTLE while acknowledging differences will remain. The most important risk factors for the primary outcome of stroke (the components of CHA2DS2-VASc score for AF stroke risk) are mostly well recorded in CPRD.[20]

There are differences in the coding systems used by the two EHR data sources and completeness of coding may differ between the two; the potential impact of this will be ascertained by comparisons of rates of diagnoses, baseline variables and prescriptions of interest. Inclusion of data source as a matching variable should prevent discrepancy between the sources from biasing results. We will explore different methods of combining Gold and Aurum, namely analysing separately by database and combining the results as a meta-analysis as an alternative to combining data before analysis.

The main focus of the study is validation of our methodology through assembling a cohort of patients comparable to the patients in ARISTOTLE and finding similar results to the trial. Criteria to determine the success of the methodology have been prespecified in the protocol. Given the use of CPRD data to determine treatment effectiveness is not yet well established, a finding that these data are not suitable to answer questions on intended effectiveness will be a useful conclusion.

### Patient and public involvement

No patient was involved.

## ETHICS AND DISSEMINATION

### Approval by ethics and scientific committees

An application for scientific approval related to use of CPRD data was approved by the Independent Scientific Advisory Committee of the Medicines and Healthcare Products Regulatory Agency (MHRA).

### Dissemination plans

The results of the study will be submitted to peer-reviewed journals and presented at conferences. Relevant charities will be contacted for guidance on dissemination of results to patients in an accessible manner. We will communicate with NICE to convey any results relevant to the guidance they have issued on AF, and with the MHRA if findings may impact the risk/benefit profile of anticoagulation treatments in patients with AF.

**Contributors** EMP, KW, IJD, UG and LS contributed to study question and design. EMP wrote the first draft of the protocol manuscript (based on the original proposal to MRC, ISAC that EMP, KW, IJD, UG and LS all contributed to). EMP, KW, IJD, UG and LS contributed to further drafts and approved the final version.

**Funding** This work was supported by the Medical Research Council through an MRC LID studentship (grant number MR/N013638/1).

**Competing interests** IJD reports grants from GlaxoSmithKline, NIHR, ABPI and MRC and holds stock in GlaxoSmithKline. LS reports grants from Wellcome, MRC, NIHR, BHF, Diabetes UK, ESRC and the EU; grants and personal fees for advisory work from GSK; and historical personal fees for advisory work from AstraZeneca. He is a Trustee of the British Heart Foundation. UG is an employee of and holds shares in GlaxoSmithKline.

**Patient consent for publication** Not required.

**Provenance and peer review** Not commissioned; externally peer reviewed.

**ORCID iDs**
Emma Maud Powell http://orcid.org/0000-0001-9427-9468
Usha Gungabissoon http://orcid.org/0000-0002-2040-1763

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
