## [Reviewer comments · BMJ Open]

ARTICLE DETAILS

TITLE (PROVISIONAL)	Real world effects of medications for stroke prevention in atrial fibrillation: protocol for a UK population-based non-interventional cohort study with validation against randomised trial results
AUTHORS	Powell, Emma; Douglas, Ian; Gungabissoon, Usha; Smeeth, Liam; Wing, Kevin

VERSION 1 – REVIEW

REVIEWER	Madeleine Durand Centre Hospitalier de l'Université de Montréal Canada
REVIEW RETURNED	28-Sep-2020

GENERAL COMMENTS	To the authors and editor, Thank you for the opportunity to revise this interesting and well written study protocol about real world effects of anticoagulants for stroke prevention in atrial fibrillation. While this subject has been much researched, authors present innovative methods. Overall the manuscript reads very clearly. I have the following comments and requests for clarifications to make : 1- Access to granular data for the landmark ARISTOTLE trial is a fantastic opportunity to study the gap between efficacy estimates obtained from RCTs and effectiveness estimates obtained from EHR cohorts studies. In objective 1, authors plan to compare the results of their ARISTOTLE-Matched cohort study to the results of the ARISTOTLE trial. This will be very interesting. I would like to see absolute event rates, in addition to hazard ratios, be compared, as this will prove crucial to understand if the observational population resembles the randomized population, or constitutes a more at-risk group. 2- In objective 1, obtaining one ARISTOTLE-Matched cohort and comparing its results to the trial will indeed be interesting, but if authors are really intent in comparing population event risk and drugs effectiveness between the randomized and EHR population, I would suggest bootstrapping this process (Objective 1, step 2 and 3). 500 or 1000 ARISTOTLE-matched cohorts could be derived, which would give us a true frequentist estimate of the rates and hazard ratios associated to exposure to warfarin vs apixaban in the EHR population. While I realize this is computationally more demanding, I believe it would be a better approach to compare observational and randomized estimates, and it would make the best use of the availability of the granular ARISTOTLE data. (This is a comment and does not require a modification of the protocol for publication)
---

3- With anticoagulation, depletion of susceptibles is a very important dynamic for bleeding outcomes. Bleeding rates decrease sharply following anticoagulation initiation, as systemic anticoagulation “unmasks” pre-existing lesions that, once anticoagulation is initiated, may start bleeding. In ARISTOTLE, randomization was stratified according to new vs prevalent users of anticoagulation. It is of paramount importance that the proportion of new vs prevalent users i) mirror that of the ARISTOTLE trial and ii) be identical in the warfarin and apixaban cohorts. I was not sure reading the text that this was the case, and would appreciate that authors clarify this section. In particular, the choice of a random date for the warfarin-treated participants is to be carefully considered; I would suggest risk set sampling to ensure similar duration of past anticoagulation for the prevalent users in the apixaban and warfarin cohorts. This is one of the very important confounding factors that would have been randomized in ARISTOTLE.

4- Objective 1 – step 1: The major feasibility challenge for this study is to derive algorithms that will identify the ARISTOTLE inclusion and exclusion criteria in the EHR. Although this in itself may require development work, if the algorithms are available now, the protocol would benefit greatly from an appendix stating those algorithms (what was used for each criteria for the feasibility counts?) For example, how will “increased bleeding risk”, “AF due to reversible causes” (a list those causes seems important in the protocol), and “conditions likely to cause death within a year” be operationalized? How many participants are expected to have available CBC, AST, ALT and creatinine values and which caliper around index date will be used to include those values? (this information in particular appears to be missing from the very helpful missing data section at the end of the manuscript.) In addition, patients with mechanical heart valves must be excluded as none will receive apixaban in the UK for treatment of AFib.

5- Objective 1 – step 2: I suggest adding to the variable list everything that is used to estimate bleeding risk (eg: all identifiable components of HAS BLED score)

6- Objective 2 - It is unclear from the protocol if patients excluded from ARISTOTLE will be pooled in a group or if stratified results will be given. I would strongly advocate for stratified results by exclusion criteria, given that some (ie: “reversible AF causes” and “no evidence of at least one additional risk factor for stroke”) represent lower-risk patients, while others (“evidence of drug/alcohol abuse” and “severe comorbid conditions”) are more likely to increase risks. Again here, absolute event rates will be of immense interest. Description of statistical methods for rates would need to be added too.

7- “Objective 3. – Objective 3 is under-developed in the protocol, and its statement in the aims and objectives (which reads: Compare treatment effectiveness between multiple individual anticoagulants in all anticoagulant recipients (no eligibility criteria other than diagnosis of AF).” Is not the same as in the body of methods, where the stratification by presence/absence of inclusion criteria to ARISTOTLE appears to be the main objective. Please revise the objective list to reflect this. Simple comparative

effectiveness would not be novel. Stratification by presence of exclusion criteria is more novel, but see comment above above different populations identified by different exclusion criterias: this needs to be carefully stratified, or interaction terms may be used in models for pre-specified potential effect modifiers, which can be different reasons for exclusion from ARISTOTLE.

Of note, as indications for Direct Oral Anticoagulants (DOACs) are more restrictive than those for warfarin (eg: renal failure, mechanical heart valves), selection bias is an important potential caveat. I would advocate against forming a cohort with “no eligibility criteria other than AF” as is stated in the objective, as this will increase this selection bias. At the very least, people on warfarin with absolute contra-indications to DOACs must be removed from the warfarin-treated cohort. Note that in this particular clinical setting, all contra-indications to DOACs increase risk of stroke, bleeding, or both.

The protocol should be clearer about cohorts constitution for objective 3 – will there be 3 matched cohorts? Figure 1 suggests there will be 2 cohorts comparing DOACs pairwise to apixaban, but the manuscript mentions warfarin. Please also clarify index date and end of follow up, matching procedures, inclusion and exclusion criteria for objective 3.

8- Methods of analysis – Using on-treatment analysis for severe bleeding will most likely result in biased results. In real life, a switch from warfarin to DOACs will most likely occur for treatment simplification or whenever DOACs can be re-imbursed, a decision that is not, most of the time, associated to a patient’s clinical status. On the other hand, a switch from DOACs to warfarin is most likely associated to occurrence of a contra-indication to DOACs (most frequently renal failure), which will also most likely increase bleeding risks. Therefore, there is selection bias due to attrition if you use as-treated exposure definitions. This would not be expected to be as important in a clinical trial, where switch from warfarin to DOAC would be less important. Our group recently published a paper about this particular dynamic in statistics in medicine, which can be found here:
<https://pubmed.ncbi.nlm.nih.gov/32812276/>

9- Adherence assessment: While adherence to warfarin will be assessed by INR measurements (a true reflection of presence of warfarin in patient’s bloodstream), adherence to apixaban will be assessed by time covered by prescriptions, which is quite « upstream » in comparison in the cascade of drug adherence: some patients will not fill the prescription, and others yet will fill it and but will not take the drugs. While there might not be any better way to assess adherence, this needs to be acknowledged. If great differences in adherence are seen between the two exposure groups, those definitions may need to be reassessed.

10- Minor comment : Direct oral anticoagulants are contra-indicated in patients with mechanical heart valves, so I suggest you remove this condition from the list when discussing the generalizability of ARISTOTLE findings : patients with mechanical heart valves are not in the population of interest we would want to generalize findings to (as opposed to people at high risk of bleeding, a sub-population we do want to generalize findings to.)

REVIEWER	Yana Vinogradova University of Nottingham United Kingdom
REVIEW RETURNED	15-Oct-2020

GENERAL COMMENTS	The submitted protocol proposes to compare results obtained from a randomised control trial with those from an observational study based on routinely-collected data. This would be an important validation study, which could demonstrate to regulatory bodies the suitability of studies using non-interventional data. The protocol is well written, the design is clearly described, and I have only minor comments and recommendations for the authors. Page 8 line 51. Study design. I would like to note that this will be a prospective study, not retrospective – the data was collected from the time of the registration of a patient with a practice contributing to the CPRD database and not at the time of an outcome. Page 10. Selection of patients. I would suggest shifting the selection time period to Jan 2014-March 2020. Although apixaban was prescribed before 2014, the number of prescriptions were very low. In one study using CPRD (BMJ 2018;362:k2505), there were only 78 patients who started using apixaban in 2012/2013 but 482 patients in 2014 (Supplementary eTable2). Having an extra 9 months at the end of the study period would also be feasible because the current version of HES covers the period up to 31 March 2020. Page 15, line 44. The same study also found well recorded SBP, with only a few patients missing this information (Supplementary eTable3). As for BMI – almost 96% of anticoagulant users had a baseline measurement for it. This information could be used to justify the exclusion of patients with missing values for SBP or BMI.
--

VERSION 1 – AUTHOR RESPONSE

#	Feedback	Response and details of any changes made
Reviewer 1		
1	Access to granular data for the landmark ARISTOTLE trial is a fantastic opportunity to study the gap between efficacy estimates obtained from RCTs and effectiveness estimates obtained from EHR cohorts studies. In objective 1, authors plan to compare the results of their ARISTOTLE-Matched cohort study to the results of the ARISTOTLE trial. This will be very interesting. I would like to see absolute event rates, in addition to hazard ratios, be compared, as this will prove crucial to understand if the observational population resembles the randomized population, or constitutes a more at-risk group.	We appreciate this suggestion and agree it would be interesting to compare absolute event rates between the ARISTOTLE trial and our EHR cohort. We have therefore made the following updates to the manuscript: Abstract Paragraph 3 line 1 “Absolute and relative risk of outcomes in the EHR trial-analogous cohort will be calculated and compared with the ARISTOTLE results;”

#	Feedback	Response and details of any changes made
		Selection of Participants Objective 1 Step 4 “The absolute rates and hazard ratio for the outcomes of interest (time to: stroke/SE, MI, major bleeding, and mortality) will then be calculated.” Statistical analysis - Methods of Analysis “All time to event endpoints will be analysed using a Cox proportional hazards model including treatment group as a covariate and prior warfarin/VKA status (experienced, naïve). Point estimates and two-sided 95% CIs will be constructed for the outcome. Absolute event rates of all outcomes of interest will also be calculated.” Statistical analysis - Validation of Results Against Aristotle Data “In order to understand the extent to which the EHR population resembles the ARISTOTLE trial population the absolute event rates of the outcomes will be compared.”
2	In objective 1, obtaining one ARISTOTLE-Matched cohort and comparing its results to the trial will indeed be interesting, but if authors are really intent in comparing population event risk and drugs effectiveness between the randomized and EHR population, I would suggest bootstrapping this process (Objective 1, step 2 and 3). 500 or 1000 ARISTOTLE-matched cohorts could be derived, which would give us a true frequentist estimate of the rates and hazard ratios associated to exposure to warfarin vs apixaban in the EHR population. While I realize this is computationally more demanding, I believe it would be a better approach to compare observational and randomized estimates, and it would make the best use of the availability of the granular ARISTOTLE data. (This is a comment and does not require a modification of the protocol for publication)	Thank you for this suggestion. As part of the study we will be exploring different methods of matching with the protocol worded to reflect this; we agree it would be interesting to consider this bootstrapping method as a way of sampling from the EHR cohort. As suggested in the comment we have not altered the manuscript but will explore this option during the analysis.
3	With anticoagulation, depletion of susceptibles is a very important dynamic for bleeding outcomes. Bleeding rates decrease sharply following anticoagulation initiation, as systemic	We agree this is an important factor from the trial to match. We had planned both to match new vs prevalent user proportions to the trial population and to match between the two treatment arms on this

#	Feedback	Response and details of any changes made
	anticoagulation “unmasks” pre-existing lesions that, once anticoagulation is initiated, may start bleeding. In ARISTOTLE, randomization was stratified according to new vs prevalent users of anticoagulation. It is of paramount importance that the proportion of new vs prevalent users i) mirror that of the ARISTOTLE trial and ii) be identical in the warfarin and apixaban cohorts. I was not sure reading the text that this was the case, and would appreciate that authors clarify this section. In particular, the choice of a random date for the warfarin-treated participants is to be carefully considered; I would suggest risk set sampling to ensure similar duration of past anticoagulation for the prevalent users in the apixaban and warfarin cohorts. This is one of the very important confounding factors that would have been randomized in ARISTOTLE.	variable – as specified via the inclusion of “Prior VKA/warfarin useexposure” as a baseline matching variable in Selection of participants Objective 1 Step 2. To further clarify this point we have added the following text: Objective 1 Step 1 “To mirror ARISTOTLE we will assess trial criteria for apixaban patients on the date of their first prescription of apixaban whilst allowing patients prescribed warfarin to become eligible at any warfarin prescription date during the study period; furthermore we will match ARISTOTLE in the proportion of new vs. prevalent users in both treatment arms.” Objective 1 Step 3 “The resulting trial matched sample of EHR apixaban patients will be matched to the warfarin ARISTOTLE-eligible EHR patients (Figure 2) using a matching method such as PSM, or CEM (final method selected based upon giving optimal sample size versus balance). Risk set sampling will be employed in order to ensure similar duration of prior VKA/warfarin exposure for the prevalent users in the apixaban and warfarin EHR cohorts. “
4	Objective 1 – step 1: The major feasibility challenge for this study is to derive algorithms that will identify the ARISTOTLE inclusion and exclusion criteria in the EHR. Although this in itself may require development work, if the algorithms are available now, the protocol would benefit greatly from an appendix stating those algorithms (what was used for each criteria for the feasibility counts?) For example, how will “increased bleeding risk”, “AF due to reversible causes” (a list those causes seems important in the protocol), and “conditions likely to cause death within a year” be operationalized? How many participants are expected to have available CBC, AST, ALT and creatinine values and which caliper around index date will be used to include those values? (this information in particular appears to be missing from the very helpful missing data section at the end of the manuscript.) In addition, patients with mechanical heart valves must be	Thank you for this comment, we agree this is an important part of the study and have added a table giving the algorithms to the extent they have already been developed to the appendix; codelists relevant to these will be included in any papers detailing the results of this study. Patients with mechanical heart valves are excluded by IE08: “If patient has medical record corresponding to a condition other than atrial fibrillation that requires chronic anticoagulation on or prior to index date then IE08=Y.

#	Feedback	Response and details of any changes made
	excluded as none will receive apixaban in the UK for treatment of AFib.	Codelist search terms include (('heart' or 'valve') and ('prosth' or 'mechanical')), 'venous thromb', and synonyms for these."
5	Objective 1 – step 2: I suggest adding to the variable list everything that is used to estimate bleeding risk (eg: all identifiable components of HAS BLED score)	Thank you for this comment, we agree the components of the HAS-BLED score should be included in the list of baseline characteristics which patients are matched on. The H (hypertension uncontrolled >160 mmHG systolic), A (abnormal liver or renal), and B (bleeding) components are exclusion criteria from the trial and are covered by the bullet point list of study exclusion criteria in Step 1. However, these H, A, and B components should be considered when matching within the patient groups excluded by the trial criteria; we have therefore amended Objectives 2 and 3 to make this clear: Objective 2 “When matching the apixaban and warfarin patients within the patient groups for this objective additional baseline variables will be considered compared with the list specified for objective 1 step 2; namely those components of the HAS-BLED score (uncontrolled hypertension, abnormal renal or liver function, and prior major bleeding or predisposition to bleeding) not included for objective 1 matching due to being ARISTOTLE exclusion criteria.” Objective 3 refers back to Obj 2 for the method to be used in matching the special patient groups therefore the above added text covers objective 3 excluded patient groups. The S (stroke history), E (elderly age), and D (alcohol consumption and use of aspirin/clopidogrel/NSAIDs) components are already in the list of Step 2.

#	Feedback	Response and details of any changes made
		Component L (Labile INR) was not included and we have therefore added this to the list of variables in Step 2:  “- Age - Sex - BMI - Systolic blood pressure - Congestive heart failure or left ventricular systolic dysfunction - Hypertension requiring treatment - Diabetes mellitus - Prior stroke/thromboembolism - Smoking status - Alcohol consumption - Level of renal impairment - Prior VKA/warfarin useexposure - Labile INR in prior users of warfarin - Concomitant use of: aspirin, antiplatelet or NSAID, lipid lowering drug therapy, or CYP3A4 inhibitor”
6	Objective 2 - It is unclear from the protocol if patients excluded from ARISTOTLE will be pooled in a group or if stratified results will be given. I would strongly advocate for stratified results by exclusion criteria, given that some (ie: “reversible AF causes” and “no evidence of at least one additional risk factor for stroke”) represent lower-risk patients, while others (“evidence of drug/alcohol abuse” and “severe comorbid conditions”) are more likely to increase risks. Again here, absolute event rates will be of immense interest. Description of statistical methods for rates would need to be added too.	Thank you for this comment – we had planned that results would be presented by separate groups i.e. not pooled and have updated the text to make this clearer: “Objective 2: we will select patient groups who would not have been included in ARISTOTLE (and therefore would not have been included in the Objective 1 cohort) or who are under-represented in ARISTOTLE. ... In these special patient populations the same outcomes as objective 1 will be assessed, with absolute and relative rates calculated separately in each special patient group.” and in Aims and objectives section: “Objective 2. Extension of trial findings: Measure treatment effects of apixaban in patient groups excluded from ARISTOTLE.”

#	Feedback	Response and details of any changes made
7	Objective 3. – Objective 3 is under-developed in the protocol, and its statement in the aims and objectives (which reads: Compare treatment effectiveness between multiple individual anticoagulants in all anticoagulant recipients (no eligibility criteria other than diagnosis of AF).” Is not the same as in the body of methods, where the stratification by presence/absence of inclusion criteria to ARISTOTLE appears to be the main objective. Please revise the objective list to reflect this. Simple comparative effectiveness would not be novel. Stratification by presence of exclusion criteria is more novel, but see comment above above different populations identified by different exclusion criterias: this needs to be carefully stratified, or interaction terms may be used in models for pre-specified potential effect modifiers, which can be different reasons for exclusion from ARISTOTLE. Of note, as indications for Direct Oral Anticoagulants (DOACs) are more restrictive than those for warfarin (eg: renal failure, mechanical heart valves), selection bias is an important potential caveat. I would advocate against forming a cohort with “no eligibility criteria other than AF” as is stated in the objective, as this will increase this selection bias. At the very least, people on warfarin with absolute contraindications to DOACs must be removed from the warfarin-treated cohort. Note that in this particular clinical setting, all contraindications to DOACs increase risk of stroke, bleeding, or both. The protocol should be clearer about cohorts constitution for objective 3 – will there be 3 matched cohorts? Figure 1 suggests there will be 2 cohorts comparing DOACs pairwise to apixaban, but the manuscript mentions warfarin. Please also clarify index date and end of follow up, matching procedures, inclusion and exclusion criteria for objective 3.	Thank you for your comments. We agree objective 3 should be focused on comparative effectiveness of the different OACs within an ARISTOTLE eligible population. We have revised the objective list to make this clear: “Objective 3. Comparative effectiveness: Compare treatment effectiveness between multiple individual anticoagulants (warfarin, apixaban, rivaroxaban, dabigatran) in ARISTOTLE-eligible cohorts and in patient groups excluded from ARISTOTLE.” Thank you for this comments, objective 3 has been reformulated to make selection be an ARISTOTLE-eligible cohort followed by comparisons within the selected excluded patient groups (see text for objective 3 below). Thank you for this feedback, we have clarified the cohorts for objective 3: as objective 3 will focus on first comparing apixaban to warfarin then comparing rivaroxaban and dabigatran to apixaban: Selection of Participants “Objective 3: we will select all patients with AF who have a prescription for apixaban, warfarin, rivaroxaban, or dabigatran in the treatment period (between 1 January 2013 and 31 July 2019). The ARISTOTLE trial criteria will be applied followed by matching the warfarin, rivaroxaban, and

#	Feedback	Response and details of any changes made
		dabigatran ARISTOTLE-eligible EHR patients in turn to the trial eligible EHR apixaban patients following the methodology outlined in Objective 1 Step 3. This process will result in the creation of 3 trial-eligible EHR cohorts: warfarin users matched to apixaban users, rivaroxaban users matched to apixaban users, and dabigatran users matched to apixaban users. Matched cohorts of excluded patient groups will also be constructed to enable pairwise comparisons of treatment effects in these groups using the method outlined in Objective 2 above. In all cohorts the same outcomes as objective 1 will be assessed with both absolute and relative treatment effects compared.”
8	Methods of analysis – Using on-treatment analysis for severe bleeding will most likely result in biased results. In real life, a switch from warfarin to DOACs will most likely occur for treatment simplification or whenever DOACs can be re-imbursed, a decision that is not, most of the time, associated to a patient’s clinical status. On the other hand, a switch from DOACs to warfarin is most likely associated to occurrence of a contra-indication to DOACs (most frequently renal failure), which will also most likely increase bleeding risks. Therefore, there is selection bias due to attrition if you use as-treated exposure definitions. This would not be expected to be as important in a clinical trial, where switch from warfarin to DOAC would be less important. Our group recently published a paper about this particular dynamic in statistics in medicine, which can be found here: https://pubmed.ncbi.nlm.nih.gov/32812276/	Thank you for highlighting this important potential source of bias. We plan to look for this bias and explore the potential impact firstly by examining the set of the patients who discontinue or switch and secondly by performing sensitivity analyses. We have added text referencing the particular problem of time-varying eligibility and plan to explore the methods referenced in the paper: Sensitivity analyses “Exclusion of patient-time post treatment discontinuation in the safety and sensitivity analyses might bias results towards a conclusion of no difference[15] and risks selection bias due to attrition [16]; the set of patients who switch or discontinue treatment will therefore be examined to ascertain whether biases of this nature may have occurred. Additional analyses may be performed using methods such as inverse-probability-of-censoring weighting (IPW) or a rank-preserving structural failure time model to estimate the treatment effect that would have been observed in the absence of treatment switching. We will explore the impact of time-varying eligibility by using methods such as a modified treatment-strategy IPW {Formatting Citation}.”
9	Adherence assessment: While adherence to warfarin will be assessed by INR measurements (a true reflection of presence of warfarin in patient’s bloodstream), adherence to apixaban will be assessed by time covered by prescriptions, which is	We agree this is a serious limitation of his study and as there is not better way to assess this measure this is listed as one of the study limitations. We have added text to cover the additional point of the different nature of the two measures:

#	Feedback	Response and details of any changes made
	quite « upstream » in comparison in the cascade of drug adherence: some patients will not fill the prescription, and others yet will fill it and but will not take the drugs. While there might not be any better way to assess adherence, this needs to be acknowledged. If great differences in adherence are seen between the two exposure groups, those definitions may need to be reassessed.	Limitations “• Adherence to medication will need to be assessed based on proxy variables (time covered by prescription for apixaban, the DOACs, time in therapeutic range based on INR measurements for warfarin); the different nature of these proxy variables means the adherence estimates may not be comparable. “ Sensitivity analyses “We will perform a supplementary analysis in patients deemed adherent (PDC ≥ 80% matching ARISTOTLE compliance limit) along with an exploratory subgroup analysis by INR TTR. The different nature of the proxy variables used for adherence in the DOACs (PDC) compared with warfarin (INR TTR) means that the adherence estimates may not be comparable; should great differences in adherence be observed between these exposure groups the definitions of adherence used may need to be reassessed. “
10	Minor comment : Direct oral anticoagulants are contra-indicated in patients with mechanical heart valves, so I suggest you remove this condition from the list when discussing the generalizability of ARISTOTLE findings : patients with mechanical heart valves are not in the population of interest we would want to generalize findings to (as opposed to people at high risk of bleeding, a sub-population we do want to generalize findings to.)	Thank you for this comment. We agree these patients should not be considered and have amended the text to remove mention of this group: Introduction – Background and rationale – para3 “The generalisability of the ARISTOTLE trial is limited by the strict eligibility criteria; evidence on apixaban’s treatment effect is therefore lacking for patients who would not have met the eligibility criteria such as those with a mechanical heart valve, at increased bleeding risk; or with severe comorbid conditions.”
Reviewer 2		
1	Page 8 line 51. Study design. I would like to note that this will be a prospective study, not retrospective – the data was collected from the time of the registration of a patient with a practice contributing to the CPRD database and not at the time of an outcome.	Thank you for comment we have clarified that this is a retrospective study using longitudinal data: Page 8 line 51. Study design “We will use a retrospective cohort study design using longitudinal data to evaluate the effects of prescribing apixaban vs warfarin and then vs other DOACs for prevention of stroke and SE in AF on key

#	Feedback	Response and details of any changes made
		effectiveness and safety outcomes using non-interventional primary care data.”
2	Page 10. Selection of patients. I would suggest shifting the selection time period to Jan 2014-March 2020. Although apixaban was prescribed before 2014, the number of prescriptions were very low. In one study using CPRD (BMJ 2018;362:k2505), there were only 78 patients who started using apixaban in 2012/2013 but 482 patients in 2014 (Supplementary eTable2). Having an extra 9 months at the end of the study period would also be feasible because the current version of HES covers the period up to 31 March 2020.	Thank you for this suggestion. We have added text (with this study referenced) that we will perform a sensitivity analysis with the selection time period shifted as suggested. In Statistical Analysis – Sensitivity analyses section text has been added: “Apixaban was a newly available drug with a low number of patients having a prescription in the first year it was available [17]; we will therefore perform a sensitivity analysis with the start of the study period shifted forwards a year to January 2014 to investigate the impact of inclusion of early adopters who may differ from later adopters of a new drug.”
3	Page 15, line 44. The same study also found well recorded SBP, with only a few patients missing this information (Supplementary eTable3). As for BMI – almost 96% of anticoagulant users had a baseline measurement for it. This information could be used to justify the exclusion of patients with missing values for SBP or BMI.	Thank you we have added text with this reference to further support exclusion of patients with missing baseline SBP or BMI: In Statistical Analysis –Missing Baseline Data section text has been added: “Furthermore, as the proportion of patients with missing baseline BMI or SBP is expected to be low (approximately 4% for BMI and <1% for SBP [17]) these patients will be excluded from the trial-eligible cohort.”
Formatting Amendments		
1	We have implemented an additional requirement to all articles to include 'Patient and Public Involvement' statement within the main text of your main document. Please refer below for more information regarding this new instruction: Patient and Public Involvement: Authors must include a statement in the methods section of the manuscript under the sub-heading 'Patient and Public Involvement'. ...	'Patient and Public Involvement' sub-heading added at the end of “Methods and analysis” section with text "No patient involved" under the sub-heading.

#	Feedback	Response and details of any changes made
	If there is no patient involved in the study, please state "No patient involved" under the sub-heading 'Patient and public involvement'.	

VERSION 2 – REVIEW

REVIEWER	Madeleine Durand Université de Montréal, Canada
REVIEW RETURNED	14-Dec-2020

GENERAL COMMENTS	Thank you for addressing all comments, I am looking forward to reading your work, this will be a most interesting study!
--

REVIEWER	Yana Vinogradova University of Nottingham United Kingdom
REVIEW RETURNED	18-Dec-2020

GENERAL COMMENTS	I am generally happy with the response and amended manuscript except for one comment. I noted in my previous review that this study is technically speaking prospective - the exposure status was ascertained before the outcome. If the authors, however, would like to downgrade their study to "retrospective" it is their choice.
---